

# Viromes of one year old infants reveal the impact of birth mode on microbiome diversity

Angela McCann[1,2,*], Feargal J. Ryan[1,2,*], Stephen R. Stockdale[1,2,3,*], Marion Dalmasso[1,5], Tony Blake[1,2], C. Anthony Ryan[1,4], Catherine Stanton[1,2], Susan Mills[1,2], Paul R. Ross[1,2,3] and Colin Hill[1,2]

[1] APC Microbiome Institute, Cork, Cork, Ireland
[2] School of Microbiology, University College Cork, Cork, Cork, Ireland
[3] Teagasc Food Research Centre, Fermoy, Cork, Ireland
[4] Department of Neonatology, Cork University Maternity Hospital, Cork, Cork, Ireland
[5] Current affiliation: Normandie University, UNICAEN, ABTE, Caen, France
[*] These authors contributed equally to this work.

## ABSTRACT

Establishing a diverse gut microbiota after birth is being increasingly recognised as important for preventing illnesses later in life. It is well established that bacterial diversity rapidly increases post-partum; however, few studies have examined the infant gut virome/phageome during this developmental period. We performed a metagenomic analysis of 20 infant faecal viromes at one year of age to determine whether spontaneous vaginal delivery (SVD) or caesarean section (CS) influenced viral composition. We find that birth mode results in distinctly different viral communities, with SVD infants having greater viral and bacteriophage diversity. We demonstrate that CrAssphage is acquired early in life, both in this cohort and two others, although no difference in birth mode is detected. A previous study has shown that bacterial OTU's (operational taxonomic units) identified in the same infants could not discriminate between birth mode at 12 months of age. Therefore, our results indicate that vertical transmission of viral communities from mother to child may play a role in shaping the early life microbiome, and that birth mode should be considered when studying the early life gut virome.

Corresponding author
Colin Hill, c.hill@ucc.ie

## INTRODUCTION

The human gut microbiota is a diverse community densely populated with bacteria, archaea, protists, fungi, and viruses. Studies focused on gut bacteria suggest that healthy individuals are characterised by high species diversity (*Heiman & Greenway, 2016*), with compositional alterations and decreased diversity linked to conditions such as obesity, diabetes and inflammatory bowel disease (IBD) (*Imhann et al., 2016*; *Karlsson et al., 2013*; *Ley et al., 2005*). Gut microbiota colonization in infants is a critical process, characterised by initial low bacterial diversity which increases with time such that by one year of age the microbiota converges towards that of an adult and fully resembles an adult microbiota
by two–five years of age (*Rodriguez et al., 2015*). Several factors have been shown to influence an infant's microbiota, from birth mode to antibiotic usage, diet, geographical location, lifestyle and age (*Milani et al., 2017*; *Rodriguez et al., 2015*). Indeed, *Hill et al. (2017)* confirmed that delivery mode and gestational age significantly influence bacterial composition in the infant gut during the first 24 weeks of life.

The gut virome is an area of growing interest with relation to the microbiota (*Breitbart et al., 2003*; *Minot et al., 2013*) and gut virome alterations have been recorded between healthy and diseased states; for instance, an increase in the taxonomic richness of *Caudovirales* has been associated with IBD (*Norman et al., 2015*). However, there is a significant knowledge gap about healthy human viral populations, with large portions of the sequence data from human virome metagenomic studies representing uncharacterised viruses either not present in current databases or described using in silico methods without host information or taxonomic assignment (*Krishnamurthy & Wang, 2017*). *Reyes et al. (2015)* demonstrated intrapersonal virome variation between adult twins over a one year period was low, while interpersonal variation was high (*Reyes et al., 2015*). *Manrique et al. (2016)* demonstrated the presence of a healthy human phageome, which is a collection a bacteriophage which are present in a large portion of healthy individuals and hypothesized that this community plays a key role in the structure of the human microbiome and by extension human health. Research on the gut virome in infancy and early life (0 –3 years) thus far has exclusively focused on longitudinal studies in twin pairs (*Lim et al., 2015*; *Reyes et al., 2015*) or has been based on a single infant (*Breitbart et al., 2008*). *Reyes et al. (2015)* identified different viral assembly stages of the gut microbiota in 20 healthy infant twin pairs (0–3 years) and revealed that this program of assembly is impaired in twins discordant for severe acute malnutrition (*Reyes et al., 2015*). *Lim et al. (2015)* conducted a longitudinal study of the virome and bacterial microbiome in four twin pairs, from birth to two years, and revealed that the expansion of the bacterial microbiome with age was accompanied by a contraction and shift in the bacteriophage composition (*Lim et al., 2015*). The work conducted by Lim et al. was unsuited to examining the impact of birth mode as only a single twin pair in that study was born by standard vaginal delivery. Reyes et al. did not report on the birth mode of the infants in their study; however, as many of the twin pairs in that study were discordant for forms of malnutrition and received a dietary intervention as a result, measuring the impact of birth mode in those infants would be complicated by confounding factors. Thus, to date, no investigation has had a study design allowing for direct investigation of the impact of birth mode on diversity and composition of the virome in early life but birth mode has been proposed as a putative modulator of microbiome diversity (*Milani et al., 2017*). Therefore, we performed a deep-sequencing metagenomic examination of faecal DNA viromes of 20 infants at one year of age and investigated whether spontaneous vaginal delivery (SVD) or caesarean section (CS) influenced gut virome composition and diversity.

## MATERIALS & METHODS

### Selection of faecal samples

Faecal samples for all infants were collected as part of the INFANTMET (*Hill et al., 2017*) study. Infant guardians were approached for written consent between February 2012 and

May 2014 in Cork University Maternity Hospital, with ethical approval provided by the Cork University Hospital Research Ethics Committee. Ethical approval reference: ECM (w) 07/02/2012. In order to control for potential variants, faecal samples were randomly chosen from those available that best met the following criteria: (1) an equal number of Spontaneous Vaginal Delivered (SVD) and emergency Caesarean-Section (CS) infants, (2) gestational-term matched infants, (3) age matched infants, and (4) a balanced number of breastfed versus bottle-fed infants from SVD and CS available samples. In addition, technical criteria included the availability of >1 g of starting faecal material. As a result, 20 infant faecal samples were selected; 10 were from SVD infants and 10 were from CS infants. Of the 10 infant faecal samples per cohort, seven and eight of the CS and SVD delivered infants, respectively, were breastfed. For details related to samples chosen in this study, see Table S1.

## Preparation of faecal viral suspensions

Viruses were separated from faecal solids using the following method. Faeces (0.5 g) was suspended in 10 ml of SM buffer (50 mM Tris-HCl; 100 mM NaCl; 8.5 mM $MgSO_4$; pH 7.5). Samples were homogenised by vortexing for 5 min, before centrifuging twice at $4,075\times$ g for 10 min at 4 °C in a swing-bucket centrifuge to remove large particulates and bacterial cells. Faecal viral suspensions were filtered twice through a 0.45 $\mu$m pore diameter filter and processed immediately for DNA extraction.

## Extraction of viral DNA

Preparation of viral suspensions and DNA extractions were optimised for the small faecal samples collected from infant adsorbent nappies. To viral suspensions, NaCl (final conc. 0.5 M) and 10 % (w/v) polyethylene glycol (PEG; average molecular weight 8,000) were dissolved before samples were chilled on ice for 3 hrs. Viruses were then precipitated from solution in a 4 °C pre-chilled centrifuge at $4,075\times$ g in a swing-bucket centrifuge for 20 min. The viral-PEG pellet was suspended in 400 $\mu$L of SM buffer, viruses were then separated from the PEG by treating the samples with an equal volume of chloroform, vortexing for 30 s, and centrifuging at $2,500\times$ g. Clarified viral preparations were treated with 20 U of DNase I and 10 U of RNase I (final concentrations; Ambion) for 1 hr at 37 °C, after the addition of 40 $\mu$L of $10\times$ Nuclease Buffer (50 mM $CaCl_2$; 10 mM $MgCl_2$). Nucleases were inactivated at 70 °C for 10 min before samples were treated with 20 $\mu$L of 10 % SDS and 2 $\mu$L of freshly prepared 20 mg/ml Proteinase K for 20 min at 56 °C. Remaining intact viruses were lysed by the addition of 100 $\mu$L of Phage Lysis Buffer (4.5 M guanidine thiocyanate; 45 mM sodium citrate; 250 mM sodium lauroyl sarcosinate; 562.5 mM $\beta$-mercaptoethanol; pH 7.0) with incubation at 65 °C for 10 min. Viral DNA was purified by two treatments with an equal volume of phenol:chloroform:isoamyl alcohol (25:24:1) and passing the resulting purified DNA through a QIAGEN Blood and Tissue Purification Kit and eluting samples in 50 $\mu$L of TE Buffer.

## Viral DNA amplification, library preparation and sequencing

Infant faecal viral DNA concentrations were equalised before amplification for sequencing using an Illustra GenomiPhi V2 kit (GE Healthcare, Little Chalfont, UK). Amplifications

of purified viral DNA were performed in triplicate on all samples as described by the manufacturer. Subsequently, an equal volume of each amplification and an equal volume of the original viral DNA purification were pooled together for paired-end Nextera XT library preparation (Illumina, San Diego, CA, USA) as described by the manufacturer. Metagenomic sequencing of stool filtrates was performed using the Illumina MiSeq (Illumina Inc., San Diego, CA, USA) by generating 300 bp paired-end read libraries following the manufacturer's instructions.

## Torque Teno Virus qPCR detection

Voided infant faeces were suspended in 1:20 (w/v) SM buffer, centrifuged twice at $4,075\times$ g for 10 min at 4 °C, before filtering twice through 0.45 µm pore diameter filters. A 200 µL aliquot of the faecal viral-enriched suspension was lysed using a QIAGEN Blood and Tissue Purification Kit following the manufacturer's recommendations with elution in 50 µL of TE Buffer. The concentration of double stranded viral DNA was calculated using a Qubit 3.0 Flurometer (Life Technologies, Carlsbad, CA, USA) using a Qubit dsDNA High Sensitivity Assay Kit (Thermo Fisher, Waltham, MA, USA). Subsequently, all dsDNA concentrations were normalised to 0.05 ng/µl. The choice of primers and conditions for detecting pan-human associated TTV by qPCR were as described by *Ssemadaali et al. (2016)*, using SensiFAST SYBR No-ROX mastermix and a LightCycler 480 thermocycler. A two-fold serial dilution of the purified TTV PCR product was also included in the qPCR for standard curve analysis.

# ANALYSIS OF VIROME SEQUENCING DATA

## Metagenomic analysis

The quality of the raw reads was visualized with FastQC v0.11.3. Nextera adapters were removed with Cutadapt v1.9.1 (*Martin, 2011*) followed by read trimming and filtering with Trimmomatic v0.36 (*Bolger, Lohse & Usadel, 2014*) to ensure a minimum length of 60, maximum length of 150, and a sliding window that cuts a read once the average quality in a window size of 4 falls below a Phred score of 30. Levels of Bacterial contamination were estimated by classifying reads with SortMeRNA v2.0 (*Kopylova, Noé & Touzet, 2012*) against the SILVA database and by aligning reads against the cpn60db (*Hill et al., 2004*) with bowtie2 in end-to-end alignment mode (*Langmead & Salzberg, 2012*). Reads were then assembled with the metaSPAdes assembler (*Nurk et al., 2017*).

Virome sequence reads were classified into known viral orders and families using the Kaiju metagenomic classifier (*Menzel, Ng & Krogh, 2016*) and the NCBI non-redundant protein database (*NCBI Resource Coordinators, 2018*). The number of *Torque Teno virus* (TTV) homologues was counted by predicting genes from all contigs with Prodigal (*Hyatt et al., 2010*), and then by BLASTp search against ORF1 from known TTV genomes (*Hsiao et al., 2016*). Prototypical crAssphage was downloaded from GenBank using accession number NC_024711.1 (*Dutilh et al., 2014*). Assembled contigs, in this study and from the Reyes et al. dataset, with similarity to crAssphage were detected by BLAST homology (*Reyes et al., 2015*). Complete, or near complete, crAss-like genomes (96–99 kb) were compared using the 'Pyani' program (https://github.com/widdowquinn/pyani), implementing the ANIm

method with a 500 bp window size. The pyani percentage identity comparison calculations were exported to R and graphed using the gplot 'heatmap.2' package. GenBank files of the 6 crAss-like phages were generated and used to visualise whole genome comparisons by EasyFig v2.2.2 (*Sullivan, Petty & Beatson, 2011*), using a minimum Blast length of 50 bp and identity of 30 bp.

## Statistical analyses

16S OTU tables from the INFANTMET (*Hill et al., 2017*) cohorts were obtained and used in this study to examine the connection between the virome and the bacteriome. In order to account for partially assembled viruses, abundances were correlated and those with a Spearman correlation of greater than 0.9 were grouped into a single feature. All statistical analyses were performed in R v3.3.0 (*R Core Team, 2016*). Alpha diversity metrics including Chao1 richness and Shannon index were computed with PhyloSeq v1.16.2 (*McMurdie & Holmes, 2013*) and plotted with ggplot2 v2.2.1 (*Wickham, 2016*). Between-group differences in alpha diversity were tested with a Mann–Whitney test (also known as a two sample Wilcoxon test). Unweighted Bray–Curtis distance was used as input for a Principle Coordinates Analysis (PCoA) as performed by the pcoa function in the ape package v4.1. Adonis tests were performed using the vegan package v2.4.3 (*Oksanen et al., 2007*) in to test community level differences. Differential abundance analyses for both virome and 16S rRNA datasets was carried out with DESeq2 (*Love, Huber & Anders, 2014*) based upon the previous reporting that it has increased sensitivity on datasets with less than 20 samples per group (*Weiss et al., 2017*).

## RESULTS & DISCUSSION

Sequencing resulted in a mean of 924,917 paired end reads per sample, which dropped to 697,558 following strict quality control, making this the deepest sequenced infant virome dataset to date (*Lim et al., 2015*; *Reyes et al., 2015*). Paired end sequence reads were classified against the nr database from NCBI using Kaiju (*Menzel, Ng & Krogh, 2016*) which translates reads into six possible read frames for classification based on amino acid homology (Fig. 1A). As with previous published findings, a large portion of sequence reads from the viromes could not be classified to any known viral taxonomic group (*Norman et al., 2015*; *Reyes et al., 2015*), with a mean 46.59% unclassified reads per sample in this cohort (Fig. 1A). For viruses which were classifiable, it was only possible to do so at higher taxonomic ranks. The most abundant viruses detected were *Caudovirales*, *Microviridae* and *Anelloviridae* (Fig. 1A), in agreement with previously published published findings (*Lim et al., 2015*). The number of sequence reads classifiable as *Anelloviridae*, a family of single stranded DNA vertebrate viruses, showed a large difference between birth modes (Fig. 1A, Wilcox test, $p = 0.02$). The *Anelloviridae* and specifically their type species, Torque Teno Virus (TTV), have been characterised by a very high prevalence in humans worldwide, although their host interaction remains poorly understood (*Spandole et al., 2015*). Previous research of the infant virome to date have described the *Anelloviridae* as important but variable members of the gut virome in the first years of life. *Lim et al. (2015)* reported *Anelloviridae* peak in abundance between six and 12 months of age

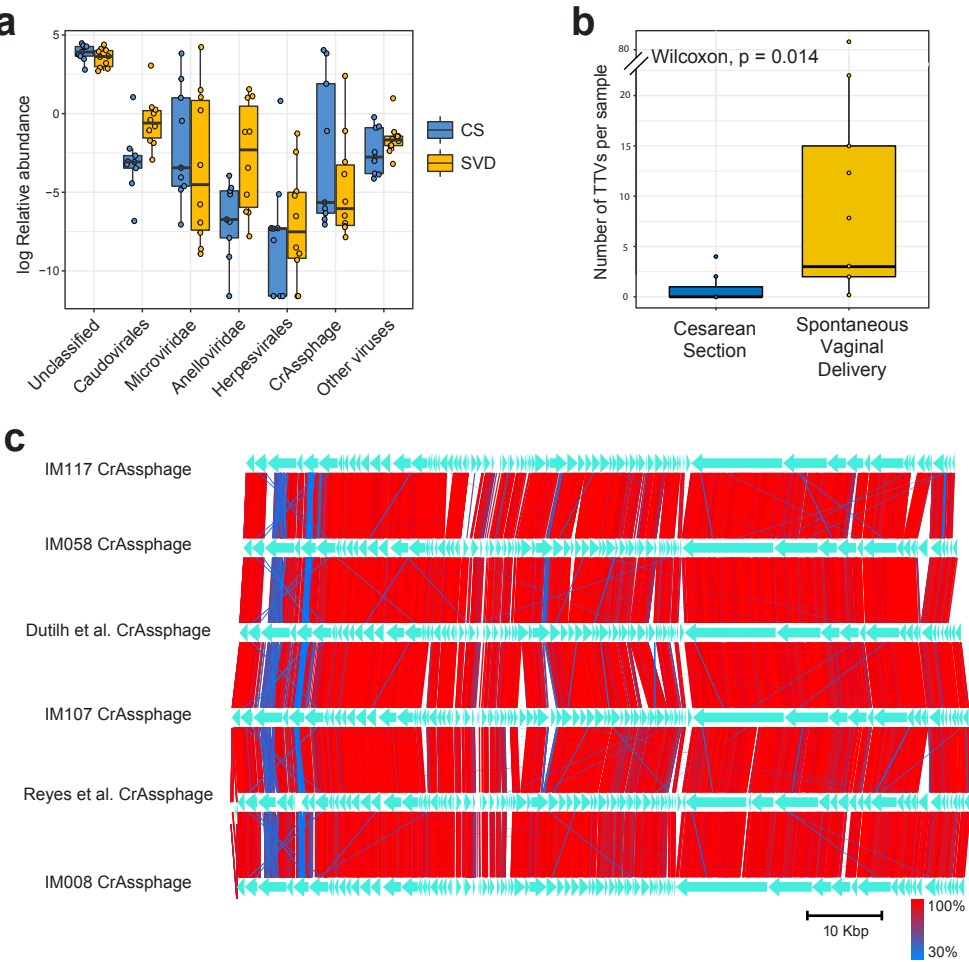

**Figure 1  Classification and abundance of known viral groups in the INFANTMET cohort.** (A) Log relative abundance of classifiable viral groups by the Kaiju amino acid classifier against the NR protein database. (B) Boxplot of the number of detectable homologues of Torque Teno Virus (TTV) ORF1 in each sample by birth mode. (C) Visualized alignment of multiple CrAssphage genomes of infant origin.

and that infants harbour multiple *Anelloviridae* species. *Reyes et al. (2015)* reported that the abundance of the *Anelloviridae* decreases after 15 months of age and that members of this family were able to discriminate twin pairs discordant for malnutrition. Further investigation of *Anelloviridae* in this cohort found that the richness of TTV was significantly increased in infants delivered by vaginal birth (Fig. 1B), but not by breastfeeding status (Fig. S1). Similar observations of vertical transmission of TTV have previously been reported, although it has been unclear whether this transmission occurs in the birth canal or in the post-partum period through mother-infant contact such as breast feeding (*Tyschik et al., 2017*). Transmission of TTV to infants could occur at any point during their development through environmental exposure, contact with other infants or parental contact. Given the ubiquity of *Anelloviridae* throughout humans (*Spandole et al., 2015*), it seems likely that transmission can happen through multiple routes and forms of contact but based
on the difference observed here it would suggest that vertical transmission is one such route. As multiple displacement amplification is known to distort the abundance of ssDNA viruses such as TTV (*Roux et al., 2016*), we sought to verify these results using quantitative PCR on the unamplified DNA from the infant faecal samples. However, due to limited sample material this was possible for only 50% of the samples (Table S2). The abundance of detected TTV DNA was found to be significantly higher in the SVD cohort over the CS group (Wilcox test, $p = 0.048$) with no difference detected by breastfeeding status (Wilcox test, $p = 0.49$).

CrAssphage is a highly abundant constituent of the human gut microbiome (*Dutilh et al., 2014*) and has previously been suggested as not present in the early life microbiome (*Lim et al., 2015*). However, we recovered several complete crAssphage genomes from infants, both in this cohort and from virome assemblies examining the gut virome of Malawian infants (Fig. 1C, Fig. S3) (*Reyes et al., 2015*). These crAssphage genomes showed high levels of nucleotide homology and synteny to the prototypical crAssphage genome as originally described by Dutilh et al. with average nucleotide identity between all six crass genomes here between 95% and 97% (Fig. 1C, Fig. S2). The impact of this highly abundant and prevalent bacteriophage on the stability of the gut microbiota is thus far unknown, but crAssphage was recently described as just one member of a previously unknown but expansive bacteriophage family (*Yutin et al., 2017*). CrAssphage is thought to predate on bacteria within the phylum Bacteroidetes (*Dutilh et al., 2014*; *Yutin et al., 2017*), which is a constituent of the infant microbiome from as early as one week after birth (*Hill et al., 2017*).

In order to assess the full diversity of the DNA virome, further analysis was based on the abundance of contigs assembled with MetaSPAdes. Estimates of 16S rRNA and 60 kDa chaperone protein (cpn60), two commonly used bacterial phylogenetic markers, and comparison to shotgun metagenomic samples from the Human Microbiome Project indicated that all samples contained low levels of bacterial contamination (Fig. S3 and Table S3). However in order to avoid any bacterial sequence being considered for analysis only contigs which passed the VirSorter virome "decontamination" mode (*Roux et al., 2015*), contained genes that corresponded to at least one known Prokaryotic Virus Orthologous Groups (pVOGs) (*Grazziotin, Koonin & Kristensen, 2016*) or showed nucleotide homology to a known virus in the nt database (*Coordinators NR, 2016*) were used for further analysis. This resulted in a total of 2,028 assembled contigs (Table S4) being taken forward for analysis, out of a possible 5,629, which recruited a median of 64.075% of reads per sample. There was no difference detected in the percentage of reads recruited between birth mode groups (Wilcox test, $p = 0.57$). Close to half of these assemblies (925, 45.1%) bore no homology at any length to anything in the nt database with an *E*-value cutoff of 1e−5, highlighting the lack of viral representation in current databases (*Krishnamurthy & Wang, 2017*). The largest assembled sequence included in the analysis a 146 kb which had a best hit of 126 bases at 88.1 percent identity to a region of Lachnoclostridium sp. YL32 annotated as a transfer RNA. Of the 2,028 included contigs only 36 were detected as circular by VirSorter (Table S4). Alpha and beta diversity analyses identified significant differences between infant viromes by birth mode (Figs. 2A, 2C, Fig. S4A). Differences in bacterial diversity at one year of age was not observed with the 16S rRNA sequencing data

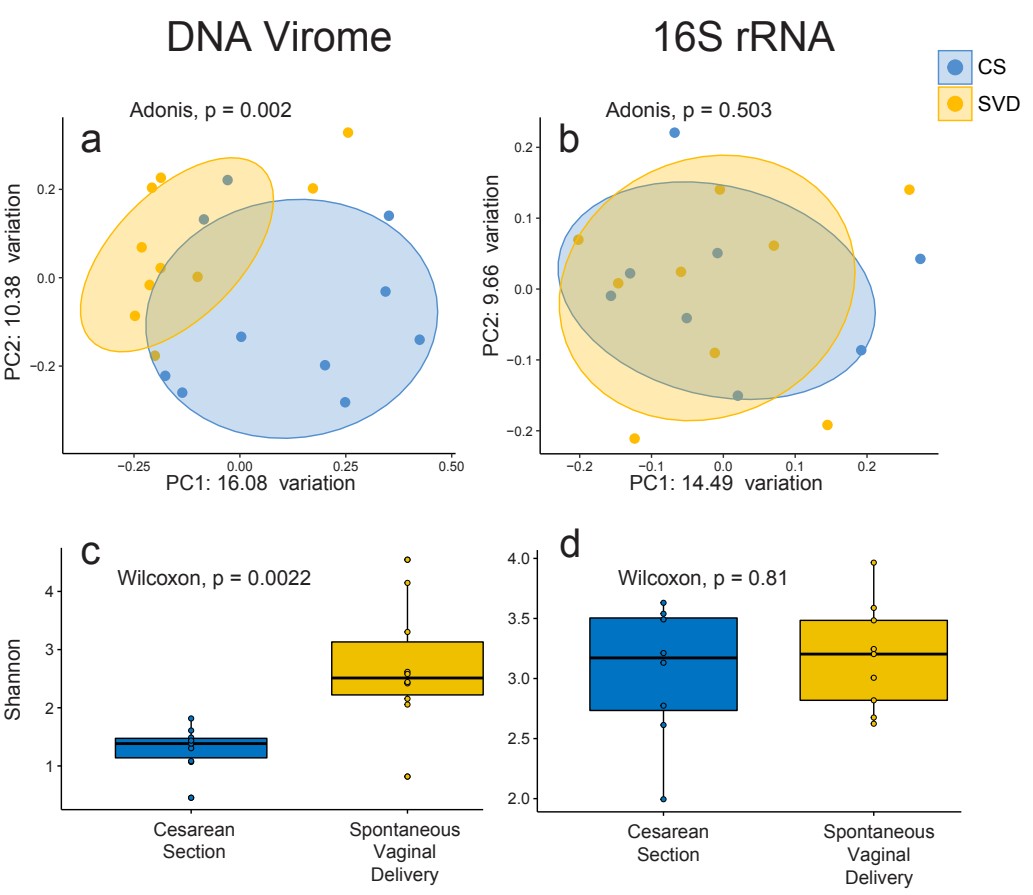

**Figure 2  Alpha and beta diversity measures for virome and 16S rRNA sequence data in the INFANTMET cohort.** PCoAs of unweighted Bray–Curtis distances for the (A) virome and (B) 16S rRNA sequence datasets, respectively. Boxplots of Shannon diversity in the (C) virome and (D) 16S rRNA sequence datasets, respectively.

(Figs. 2B, 2D, Fig. S4B). The lack of taxonomic resolution with the 16S rRNA gene possibly masks diversity differences at the species, or strain level which may only be observable through shotgun metagenomic sequencing (*Yarza et al., 2014*).

No single virus, or viral taxon, was identified as being universally absent in CS and universally present in SVD. However, DESeq2 did identify 32 contigs differentially abundant by birth mode, including TTV and several contigs bearing high levels of nucleotide homology to *Bifidobacteria* temperate phages including those from *Bifidobacterium longum* subsp. *infantis* and subsp. *longum* (Table S5) being increased in infants born by SVD. This may be reflective of differential colonisation of *Bifidobacterium* by birth mode, an observation which is supported by 16S rRNA sequence based studies (*Hill et al., 2017*). Only 5 of the differentially abundant contigs were significantly increased in CS relative to SVD, none of which showed high enough levels of homology to reliably infer their taxonomy or host (Tables S4 & S5).

## CONCLUSION

Birth mode has been established to impact the microbiome but the exact mechanism or duration of the impact has yet to be established. Here we observe a strong correlation between birth mode and diversity of the gut virome at one year of age. This may indicate that vertical transmission of viral communities may help shape the early life microbiome. In theory, the ability of the virome to predate on bacterial hosts could increase bacterial diversity, and thus assist overall community fitness. However, before causation can be established this phenomena will need to be characterized both in animal models and in larger human cohorts incorporating longitudinal sample collection. Future studies of gut virome composition and diversity in the first years of life should also consider birth mode as a potential confounding factor.

### Funding

The APC Microbiome Institute is a research centre funded by Science Foundation Ireland (SFI), through the Irish Government's National Development Plan (Grant Number 12/RC/2273). The INFANTMET project was funded by the Irish Department of Agriculture, Food and Marine. The funders had no role in study design, data collection and analysis, decision to publish, or preparation of the manuscript.

### Grant Disclosures

The following grant information was disclosed by the authors:
Science Foundation Ireland (SFI): 12/RC/2273.
Irish Department of Agriculture, Food and Marine.

### Competing Interests

The authors declare there are no competing interests.

### Author Contributions

- Angela McCann analyzed the data, contributed reagents/materials/analysis tools.
- Feargal J. Ryan analyzed the data, contributed reagents/materials/analysis tools, prepared figures and/or tables, authored or reviewed drafts of the paper, approved the final draft.
- Stephen R. Stockdale performed the experiments, analyzed the data, contributed reagents/materials/analysis tools, prepared figures and/or tables, authored or reviewed drafts of the paper, approved the final draft.
- Marion Dalmasso performed the experiments, contributed reagents/materials/analysis tools.
- Tony Blake analyzed the data.
- C. Anthony Ryan, Catherine Stanton and Paul R. Ross conceived and designed the experiments.
- Susan Mills authored or reviewed drafts of the paper.
- Colin Hill conceived and designed the experiments, authored or reviewed drafts of the paper.

## Human Ethics

The following information was supplied relating to ethical approvals (i.e., approving body and any reference numbers):

The infants included in this study are part of the INFANTMET study cohort. Infant guardians were approached for written consent between February 2012 and May 2014 in Cork University Maternity Hospital, with ethical approval provided by the Cork University Hospital Research Ethics Committee.

## DNA Deposition

The following information was supplied regarding the deposition of DNA sequences:

Sequence Read Archive:

PRJNA385126.

## Data Availability

The raw sequence data has been deposited in the NCBI Sequence Read Archive under the accession number SRP106048. Accession numbers for each individual subject are available in Table S1. FastQC reports, R code, assembled sequences, BLASTn output against the nt database, taxonomic assignments 16S sequences and count tables for both virome and 16S rRNA data were deposited in FigShare and are available at Ryan, Feargal (2018): INFANTMET Virome Study. figshare. Fileset. https://doi.org/10.6084/m9.figshare.5948443.v1

## Supplemental Information

Supplemental information for this article can be found online at http://dx.doi.org/10.7717/peerj.4694#supplemental-information.

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
