# Peer review of "Viromes of one year old infants reveal the impact of birth mode on microbiome diversity"

_PeerJ, doi:10.7717/peerj.4694_

## Round 0.1 · original submission · Major Revisions

· Academic Editor

Major Revisions

While all of the reviewers agreed that this is an interesting study, they raise several major concerns that should be addressed in a potential revision. In particular, please pay attention to the comments of Reviewer #3 who noted several places where data (and accompanying statistics) that are critical for supporting the paper's conclusions are not shown. Given the well-known bias of Genomiphi amplification towards small circular genomes, I agree with this reviewer that additional controls (ideally qPCR before genomiphi) would be needed to make claims regarding abundance of TTV. It's not sufficient to just say that the samples were processed in an identical fashion because the amount of bias depends on things like the complexity of DNA molecules in the sample and the initial amount of circular DNA, and thus can be highly variable.

In addition, Reviewer #2 raises some important concerns regarding details of the methods that should be included, and please enhance the discussion on the differing data on anellovirus colonization from the Lim et al. paper.

Finally, Reviewer #1 asks for a definition of what is considered "early life". This is a crucial definition for reaching the conclusions in your paper, and for other studies to interpret your results in subsequent research.

·

Basic reporting

no comments

Experimental design

The manuscripts question is well defined, relevant and interesting. An important point that is made throughout the manuscript is regarding the "early life virome/microbiome", however, is not well defined anywhere in the manuscript what is considered as "early". The introduction mentions that individuals at 1year of age begin to resemble an adult microbiome, so intuitively it can be thought that an "early" microbiome can be defined as the first weeks of months of age. However, they use samples at 1 year of age to define differences in the "early" microbiome, where many changes due to environmental exposure could have significant effects.

2) Line 174 mentions the presence of crassphage in that early life microbiome. Due to the same argument as before, unless is well defined what "early" represents, and the exact age of the individuals at the sampling where the crassphage was identified, is difficult to make the point about crassPhage being present at an early age.

3) Line 52: cites Reyes et al, 2010 as a reference of the "early virome", however, that manuscript describe the differences in virome between adolescent/adults twins, samples taken within a 3 year period, not at 0-3 years in age. This should be corrected and any reference to those results as representing an early age virome should be revised.

4) The earliest study of the gut virome is: Viral diversity and dynamics in an infant gut by Breitbart et al in 2008. Should be cited.

Regarding the Methods:
5) What was the insert size of the library? If less than 500 the pair end reads could be assembled before a de novo assembly. Why the trimming had a max length of 250 if 300bp were generated? Important data may have been lost.

6) Line 125, why the prediction was only done for those lengths? Sometimes a partial assembly may lead to smaller contigs, but still real, thus, ignoring those may lead to an underestimation of the diversity.

7) Line 141, check the sentence “All statistical analyses R v3.3.0"

8) Line 166: Mentions richness and abundance of TTV and references Fig 1b, but the figure only shows Number of TTVs per sample (richness) so no abundance is shown.

9) No info about assembly statistics is given, number of assembled contig, average length or coverage, etc.

10) No mention of the differences in identity about the crassphage is done either (Fig S3), which is cited in line 176 but no mention of the results shown or discussion about those is done.

Validity of the findings

The only comment here is following the comment on defining an "early virome" and the fact that the samples were originated from ~1 year old individual. Line 170 states that the fact that breast-feeding is not a factor determining the TTV abundance, "strongly supporting transfer during vaginal delivery". I consider that during 1 year of age there could be many other environmental factors that could be relevant for that TTV colonization that need to be ruled out, or stronger evidence for the vaginal delivery transfer is needed to support such statement.

Additional comments

Three minor comments:
Fig S1: A log or square root transformation will be ideal so the outliers don’t take most of the image.

No legend for supplementary figures was found.

Fig S3 is cited before S2.

Reviewer 2 ·

Basic reporting

The basic reporting is satisfactory.

Experimental design

The research is primarily exploratory and hypothesis generating. The methods are adequately described.

Validity of the findings

The results on colonization with anelloviruses differ from a published paper (Lim et al.). This deserves more discussion.

Additional comments

The authors present a study of the virome in stool of 20 one year old infants. They find groups of viruses typically seen in human stool, including Caudovirales, Microviridae, Anelloviridae, and others. The authors obtain data indicating that TTVs (from the Anelloviridae group) may be enriched in babies with spontaneous vaginal delivery versus cesarean section. Samples from spontaneous vaginal delivery are also proposed to be more diverse. The authors thus propose that verticle transmission is important in formation of the virome at one year.

The formation of the virome early in life is an interesting question, and the work is generally technically well done. The paper would be improved by addressing the following points:

1. Data in this paper differs from that of Lim et al. in suggesting TTVs are inherited vertically. Lim et al. studied samples taken earlier, and saw TTVs were not present in the first time point for most subjects, but were acquired later. Unfortunately the authors’ only sample at one year, so the timing of TTV acquisition is unclear, and the main point is in real doubt. This should be discussed carefully.

2.The authors were not clear about whether they analyzed negative controls (blank virome purifications), and if they did what sequences were in them. The issue of DNA contamination in low biomass samples is increasingly recognized as a major issue in the field. Please specify.

3. For the diversity analysis, was this done with database attributions, or over all contigs? How was possible fragmentation of viral genomes handled? That is, a large viral genome, partially sequenced, could potentially yield many short contigs—how was this dealt with?

Reviewer 3 ·

Basic reporting

This short paper reports on viromes from feces of 20 children at 1 year, 10 of which were born by caesarean section, 10 by spontaneous vaginal delivery. The sequencing depth is high, apparently many contigs could be assembled from the viromes. There seems to be some differences in the virome composition with the birth mode, which would be interesting. At present however, results need to be strengthened to reach any conclusion.

Experimental design

Two examples were results are not reported fully:
* Lane 121: “Reads were then assembled with the metaSPAdes assembler”. Results not shown. How many contigs of what length? How many reads are mapped back on these contigs? No abundance table given, no description of the contigs. Some phages of the study are said to infect Bifidobacteria, a genus for which 60 prophages have been described (Lugli et al. Env Microbiol 2016), so details about these should be given.

* Lane 123: "Raw reads were classified using Kaiju": results not shown.

Concerning the risk of contamination of viral DNA with bacterial material: suppl. table 2 shows the 16S DNA content of the viromes, which in 3 cases is quite high (above 0.02%). This means that prophages could confound the signal of the temperate phages. Appropriate controls should therefore be added when quantifying viral contigs of known bacterial hosts (such as Bifidobacteria infecting phages), by mapping reads on bacterial DNA of Bifidobacterium.

Presentation of the results needs to be improved: all data are shown as boxplots, but given the small number of data points, all points should be shown instead, together with a bar for the average.

The sequences have been deposited, but have not yet accession numbers.

Validity of the findings

The two main results of this manuscript need to be strengthened:
1. Lanes 160 -166: A difference in abundance is suspected for TTV, a circular single strand DNA virus. Authors are aware of the distortions introduced by genomiphi amplification, but nevertheless consider that “all samples were processed in an identical fashion allowing comparisons between groups”.

This should be proven with repeats, or controls. A qPCR confirmation on the virome samples prior genomiphi amplification would be a good way to confirm this interesting result.

2. Lane 192: “However, DESeq2 did identify 32 contigs differentially abundant by birth mode, including TTV and several contigs bearing high levels of nucleotide homology to Bifidobacteria prophages.”

Results are not shown. Are the differences statistically significant? Which Bifidobacteria prophages were found (technically, they should be named temperate phages rather than prophages. ?


Lane 196 :”While this study only analysed 20 infants it is sufficiently powered to provide the first evidence that birth mode significantly affects the infant gut virome. Given the differences in viral diversity observed here, we hypothesize birth mode plays a role in shaping the early life gut microbiome and that the virome is an effective biomarker of bacterial diversity at sub-OTU level”

This conclusion is not supported by the data.

Additional comments

Most of the data are not shown, statistical significance is not always given and the conclusions are therefore not supported by the data. So I cannot recommend publication of this manuscript at this premature stage.

---

## Round 0.2 · Minor Revisions

· Academic Editor

Minor Revisions

Thank you for your attention in addressing the reviewers' concerns.

Please make the final changes recommended by Reviewer #1 and Reviewer #3 and I will accept the revised manuscript.

·

Basic reporting

No comment

Experimental design

No comment

Validity of the findings

No comment

Additional comments

The manuscript has improved from the original version and all my concerns have been adequately addressed. My only concern remaining is the suggestion of causation in the abstract regarding mode of birth and the impact on the virome. Definitely there is a strong correlation and there are clear and statistical differences in the virome depending on the birth mode. However, even though other factors were taken into account, the fact that there is only one time point and is after 1 year of age, I don't think there is enough evidence for concluding causation.

Reviewer 2 ·

Basic reporting

Fine

Experimental design

Fine

Validity of the findings

Fine

Additional comments

Fine

Reviewer 3 ·

Basic reporting

OK

Experimental design

OK

Validity of the findings

OK

Additional comments

Thanks for this new version, where more data are given.
The qPCR data on TTV really add strength to your conclusions.
I have only two points left:
(1) I really insist that the box plots should be replaced by graphs with dots, with a bar for the median. This is not aesthetics, but shows the real data points.
(2) A mistake is present in the following new sentence “CrAssphage is thought to predate on bacteria within the genus Bacteroides (Dutilh et al 2014)”. Replace "genus Bacteroides" by "phylum Bacteroidetes" and add the Yutin et al reference, confirming this possible origin (a very large branch in fact).

Once these two things are corrected, I accept publication,

---

## Round 0.3 · accepted · Accept

· Academic Editor

Accept

Thank you for taking the time to make all the edits requested by the reviewers and congratulations on an excellent manuscript!

#